# Research on the Precision Measurement Method of Flat Screen Gap Based on Mobile Vision

Xuhui Ye [1], Fusheng Wang [1], Qianyou Yang [2], Xinyu Hu [1,*], Junfeng Meng [1] and Lian Song [3]

[1] School of Mechanical Engineering, Hubei University of Technology, Wuhan 430068, China; yxh89@hbut.edu.cn (X.Y.); 102110162@hbut.edu.cn (F.W.); 18838543577@163.com (J.M.)
[2] Chengchao Co., Ltd. of WISCO Resources Group, Wuhan 430073, China; ezccyqy@163.com
[3] Chongqing Research Institute, Wuhan University of Technology, Chongqing 401151, China; songliang6543@163.com
* Correspondence: 19991012@hbut.edu.cn

**Abstract:** In view of the problem that the measurement accuracy of screen gaps is affected by the R region of the transition rounded corner of the large-size flat back cover, a mobile vision system with dual linear motor synchronous drive is designed to realize precise positioning of monocular vision in 16 local image acquisition positions set in the R region. Full-size measurement of the R region is achieved using local information fusion to accurately measure the screen gap. First, to solve the problem of edge breaking and the blurring of local images, a region extraction and segmentation method based on block statistics is proposed to quickly obtain accurate outer edges. Then, an adaptive Hough line detection method based on dichotomy is proposed to accurately locate the outer edges. The experimental results show that the accuracy of region extraction and segmentation is up to 99.68% within 60 ms; the angle error and position error of line detection are less than 0.01° and 0.2 pixels, respectively. With optimal anti-interference performance, the proposed algorithm realizes the adaptive and accurate positioning of the outer edge within 200 ms; the measurement time and the measurement error of the measurement system are less than 6.0 s and 0.03 mm, respectively.

**Keywords:** large size; mobile vision; pixel statistics; region extraction and segmentation; straight line detection

## 1. Introduction

Screen gap refers to the small space between the touch screen and the back cover of a flat device during the assembly process. The dimensional tolerance of the screen gap plays a critical role in determining the overall product quality. Although the touch screen is manufactured with high precision, the flat back cover often suffers from dimensional deviations due to inadequate manufacturing processes. The R region of the flat back cover refers to a small rounded corner area specifically designed to facilitate the assembly process for touch screens. The size of this R region directly influences the dimensional tolerance of the screen gap. At present, manual plug gauges are mostly used to measure the screen size, which proves to be an inefficient method. Moreover, the measurement results obtained through manual plug gauges are subjective and lack the ability to ensure measurement accuracy. With advancements in precision measurement technology, the precision measurement of the screen gap can be indirectly realized through non-contact and high-precision measurement. Laser measurement and visual measurement are two widely adopted technologies [1–3] in achieving precise measurement in this context.

The laser measurement system utilizes a laser to project onto the measured surface and calculate the size of the object based on the obtained surface information. Liu et al. proposed a measurement system consisting of three parallel lasers to address the issue of measurement accuracy when small targets, such as tilted cracks, are involved [4]. Wang et al. developed a spot laser scanning system reflected by a right-angle prism to achieve precise

measurement of the raceway profile of the ball nut by analyzing the error of the laser and the prism [5]. Liu et al. employed a linear laser triangulation ranging sensor to capture the height point cloud data of the ring gauge under measurement. The experimental results demonstrated that the measurement error for the circular hole with a diameter of 3 mm was as low as 2 um [6]. Mei et al. improved measurement accuracy and data stability by offsetting the laser beam at a specific distance from the center of the gear. They achieved this by establishing an optimal offset model and measuring the tooth profile at the calculated optimal position [7]. Shi used a line laser sensor to collect information on the rotating gear, and the measurement deviation of the whole tooth surface of the gear reached the micron level by establishing a measurement model of the spatial coordinate system [8]. However, laser sensors can be costly, and the accuracy of laser measurements on complex surfaces such as flat back covers will be lowered compared with visual measurement methods.

In vision measurement systems, light sources are typically arranged according to specific scenes. A monocular vision system is enough to measure the size of plane objects by using image processing and coordinate conversion. Wu et al. conducted a vibration displacement measurement of a vertical rope by mounting a high-speed camera on a tripod. They employed edge detection techniques on the inter-frame differential image of the rope to accurately measure the displacement caused by the vibration [9]. Hao et al. studied a measurement system for rotating lines and achieved high-precision angle measurements by enhancing the LSD algorithm [10]. Hu et al. mounted the camera onto an adjustable bracket and captured a clear image of the O-ring using the front and rear lighting LEDs. By employing three spline interpolations [11], the measured outer diameter error remained within 0.1 mm. Li et al. positioned the back-illuminated workpiece on a rotary table to facilitate the dimensional measurement of the groove of the inner wire joint. The average measurement time for each groove was less than 0.4 s [12], indicating a fast and efficient measurement process. Zhao et al. introduced a precision visual measurement system for micro-small optical components by proposing a dark-field scattering illumination scheme that combined dome light and coaxial light [13]. Jiang et al. utilized a vision measurement system comprising a CMOS camera and coaxial telecentric parallel light source. They proposed a high-precision measurement method of thread parameters based on contour corner detection [14].

According to the above study, the precision measurement method based on global imaging usually collects a high-resolution image of an object in a relatively fixed small field of view and then calculates the size of the object through the algorithm. This method is only suitable for the precision measurement of small-sized objects due to the limitation of the size of the image sensor. When measuring large-sized objects, the field of view has difficulty covering the entire object. Even if the entire object can be covered, the limitation of pixel accuracy will reduce the accuracy of the measurement. Especially for the telecentric lens commonly used for precision measurement, it is difficult to obtain a larger field of view and higher pixel accuracy at the same time. Therefore, the traditional method cannot complete the measurement task or achieve high-precision measurement when measuring large-size objects.

In this paper, the linear motor module and the telecentric monocular vision system are combined into a mobile vision system. The um-level high-precision positioning of the linear motor and the low distortion characteristics of the telecentric camera are used to realize the precision measurement of the vision system in a large working range, which solves the problem where the traditional method could not measure the large-size object accurately. Based on the mobile vision system, in order to increase the effectiveness of the actual screen gap measurement process and take into account the accuracy of the measurement, a large-scale object precision measurement method using fusing system positioning information and multiple local images is suggested. In order to resist interference and defects in local images and properly locate the outer edge of the screen gap, a quick region segmentation algorithm is proposed based on pixel statistical theory, and an adaptive line detection algorithm is created by combining dichotomy and Hough transform.

The remainder of this article is structured as follows: The structure of a mobile vision system and the principle of full-size measurement are elaborated in Section 1. Section 2 presents the methods of region segmentation and adaptive line detection. Experimental data and measurement results are interpreted in Section 3. In Section 4, we discuss the research of this paper. Finally, our conclusions and future works are summarized in Section 5.

## 2. Structure Design and Measurement Principle of Mobile Vision System

### 2.1. Structure Design of Mobile Vision Measurement System

The vision system and the mobile system form the basis of the mobile vision measuring system. With the use of a high-resolution camera, the vision system displays a crisp image of the thing being measured. To accomplish precise mobile positioning, the mobile system drives the visual system simultaneously using high-precision dual linear motors. The mobile vision measurement system is shown in Figure 1.

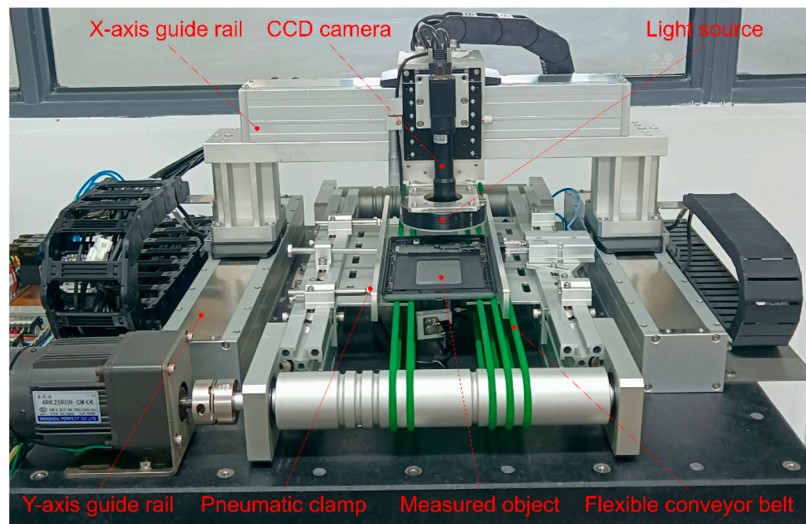

**Figure 1.** Mobile vision measurement system.

The MV-CA050-A0GM black and white CCD camera with MVL-HT-1-65 telecentric lens was chosen for its vision system due to its high standards for picture quality in precision measurement, and the P-RV-5080-1 annular light source was installed on the front. A vision system with a resolution of 2448 × 2048 can deliver sharp images.

The entire gantry framework was used for the mobile system. The driving motor is a Hanqu HWP220 high-precision linear motor, and the Siemens S7-1200 PLC manages the two motors to drive the x and y axes simultaneously. To precisely place the vision system in a two-dimensional plane and increase the working distance of monocular vision to 340 × 340 mm, the moving distance was fed back by the grating ruler.

The relevant information of the hardware is shown in Table 1.

**Table 1.** Hardware information of the system.

| Hardware | Manufacture | Country |
| --- | --- | --- |
| camera | HIKROBOT | China |
| lens | HIKROBOT | China |
| light | Jiali | China |
| linear motor | Hanqu | China |
| PLC | Siemens | Germany |

### 2.2. Principle of Full-Size Measurement Based on Mobile Vision

The left, lower, right, and upper portions of the R region of the flat cover were designated as *L*1, *L*2, *L*3, and *L*4, respectively, according to the flat cover's rectangular con-

struction. Through the workpiece coordinate system, 5, 3, 5, and 3 local image acquisition positions were, respectively, set at *L1*, *L2*, *L3*, and *L4* at equal intervals, as shown in Figure 2. The midway of the outer margins of the local image in the R region was chosen as the measuring point because the center distortion of the CCD camera was minimal during imaging [15], and the physical coordinates of the measuring point were acquired from the internal and external characteristics of the calibrated camera.

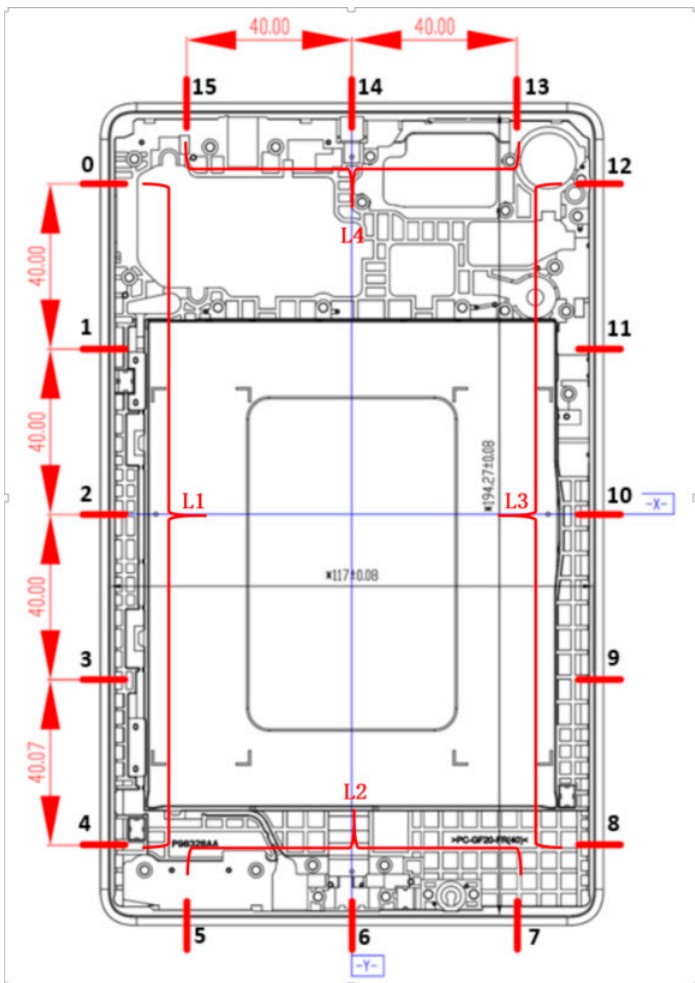

**Figure 2.** Total of 16 local image acquisition locations.

Let us assume that the physical coordinate $P_i$ of the measurement point is obtained at the local image acquisition location of *L1*, and the physical coordinate $P_j$ of the measurement point is obtained at *j* where the vision system is located at the local image acquisition position corresponding to *L3*. The distance to the local outside edge of the R region can be calculated using the global coordinate system. The schematic diagram of the distance measurement of the local outer edge of the R region is shown in Figure 3.

Let the moving distance of the vision system on the x and y axes fed back by the grating ruler be $\Delta x_{i-j}$ and $\Delta y_{i-j}$, respectively. $\Delta y_{i-j}$ is not 0 considering the error of the moving system. The outer edge of the screen gap is not parallel to the *y* axis of the mobile system due to the clamping error. To determine its angle, we take the median of the outer edge straight line detection angle across 16 local images. From the schematic diagram of the local outer edge of the R region, it can be seen that the distance $D_{i-j}$ of the measurement point to the local outer edge of the R region can be calculated according to Equation (1). Calculating the distance of the ranging results of the local outside edge of each group yields

the distance of the outer edge of the R area. The size of the screen gap is inferred from the dimensions of a conventional touch screen.

$$\overline{P_iP_j} = \sqrt{\left(P_j.y + \Delta y_{i-j} - P_i.y\right)^2 + \left(P_j.x + \Delta x_{i-j} - P_i.x\right)^2}$$

$$D_{i-j} = \overline{P_iP_j} \times \cos\left(\varphi - \arctan(\Delta y_{i-j}/\Delta x_{i-j})\right)$$

(1)

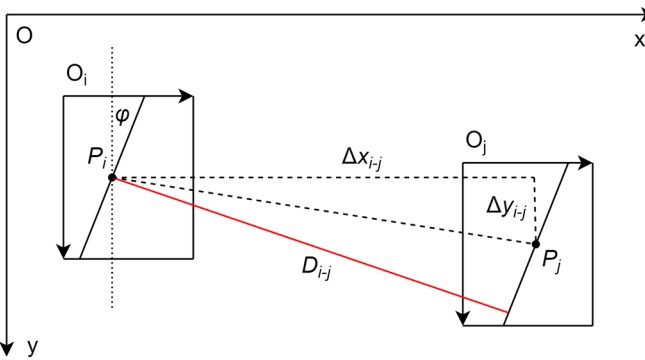

**Figure 3.** Schematic diagram of distance measurement at local outer edge of R area.

A programmable logic controller (PLC) and two synchronous motors are used to drive the mobile system and complete positioning in the workflow of the mobile vision measuring system. In order to acquire the local picture in the R region, the vision system first locates the acquisition position consecutively. The PC host computer then employs multithreading technology to process the image concurrently. Second, the camera's calibrated internal and exterior characteristics are used to determine the physical coordinates of the measuring point after the image algorithm recognizes the line on the edge and uses its midpoint as the measurement point. Third, the PLC feeds back the positioning information to the PC host computer by obtaining the moving distance of the grating ruler. The local information fusion is then carried out to complete the measurement together with the positioning of the vision system and the physical coordinate information of the local image acquisition location measurement point. Figure 4 depicts the mobile vision system's workflow.

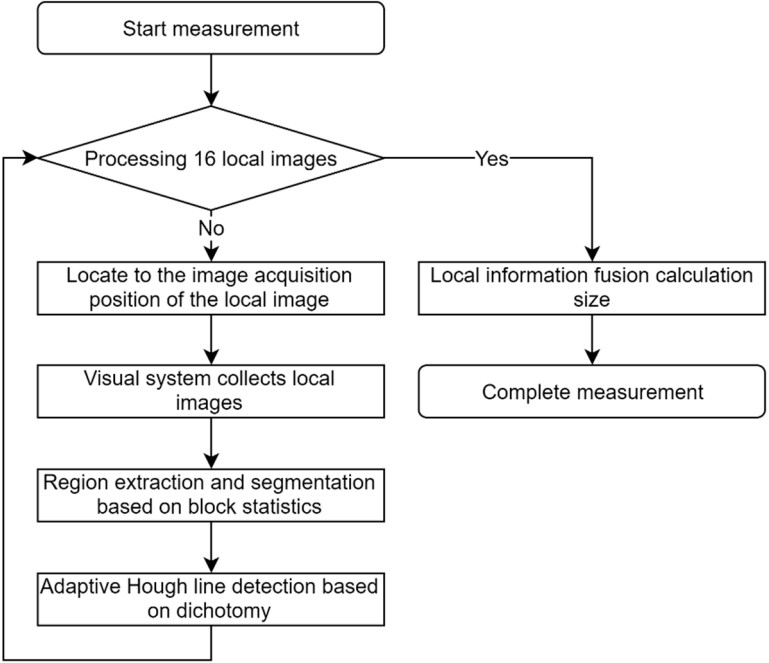

**Figure 4.** Workflow of the mobile vision system.

### 3. Precise Measurement of Screen Gap Based on Outer Edge Detection of the Flat R Region

By measuring the distance of the outer edge of the R region in full size, it is possible to indirectly achieve the accuracy measurement of the screen gap. The main challenge for the image algorithm in the measurement system is to realize the adaptive and accurate line detection of the outer edge of the R region. In the current common line detection methods, LSD line detection may reliably detect local small edges in the currently used methods for line identification [16], but many local line detection results make it challenging to discover the overall edge. Hough line detection has optimal line identification ability in an interference environment, but it requires the manual setting of an appropriate accumulator threshold [17]. LS straight line fitting has higher precision and a faster speed, but it depends on a large number of accurate fitting points [18].

The workpiece surface has surface flaws such as scratches, bright spots, and an uneven gray scale, all of which cause localized damage and blurring of the edge in the image while performing linear detection on the outer edge of the R region. It is difficult to meet the requirements of automation and high-precision measurement when using a two-dimensional measuring instrument to detect the outer edge of the R region. To illustrate the technicality and practicability of the algorithm in the same research, a technical route for the algorithm of straight line detection outside the R region of the flat back cover is proposed, as shown in Figure 5.

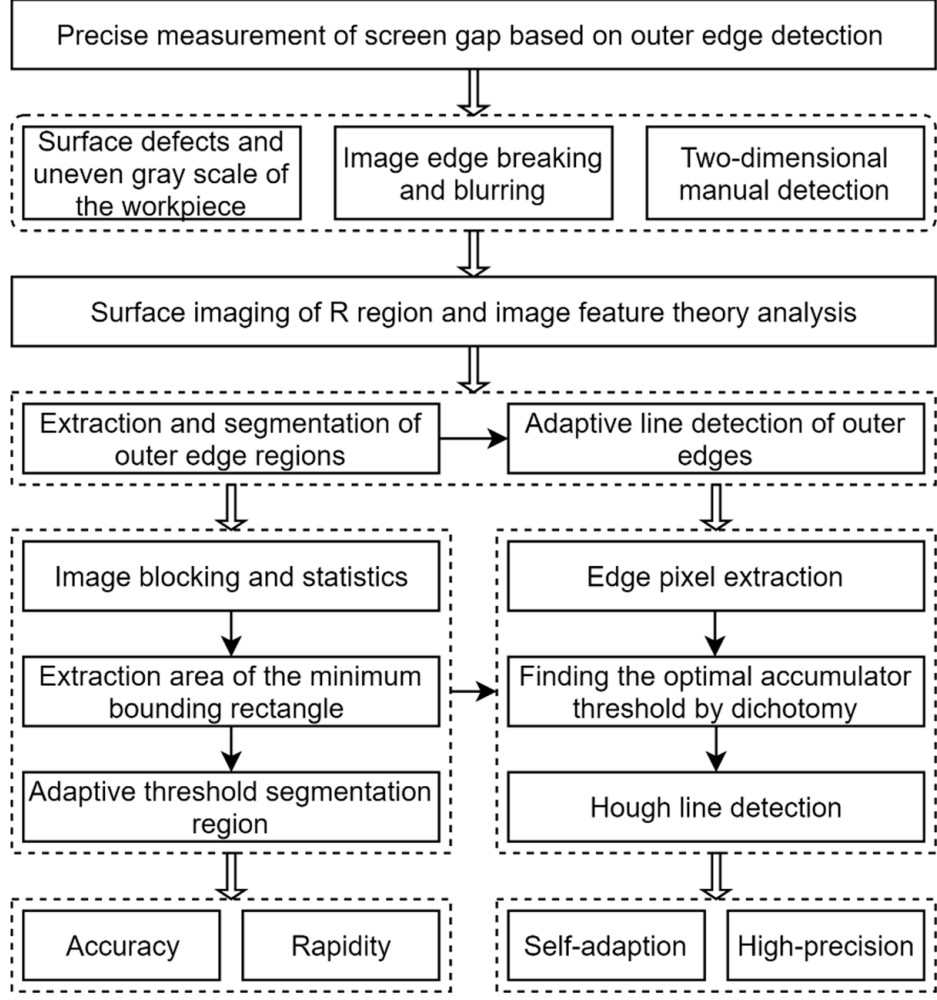

**Figure 5.** Technical route of the algorithm.

### 3.1. Theoretical Analysis of Surface Imaging and Image Characteristics of the Flat R Region

For the R region of the inverted transition rounded corners on the surface of the flat back cover, the area different from the R region is designated as A1, and the intersecting R region is A2. Due to the structural difference of the surface of the flat back cover, the intensity and angle of the reflected light in different areas are different [19]: The highest reflected light intensity and smallest reflected light deviation are found in A1, whereas the lowest reflected light intensity and greatest reflected light deviation are found in the R area. Figure 6a depicts the flat rear cover's surface makeup and the characteristics of reflected light.

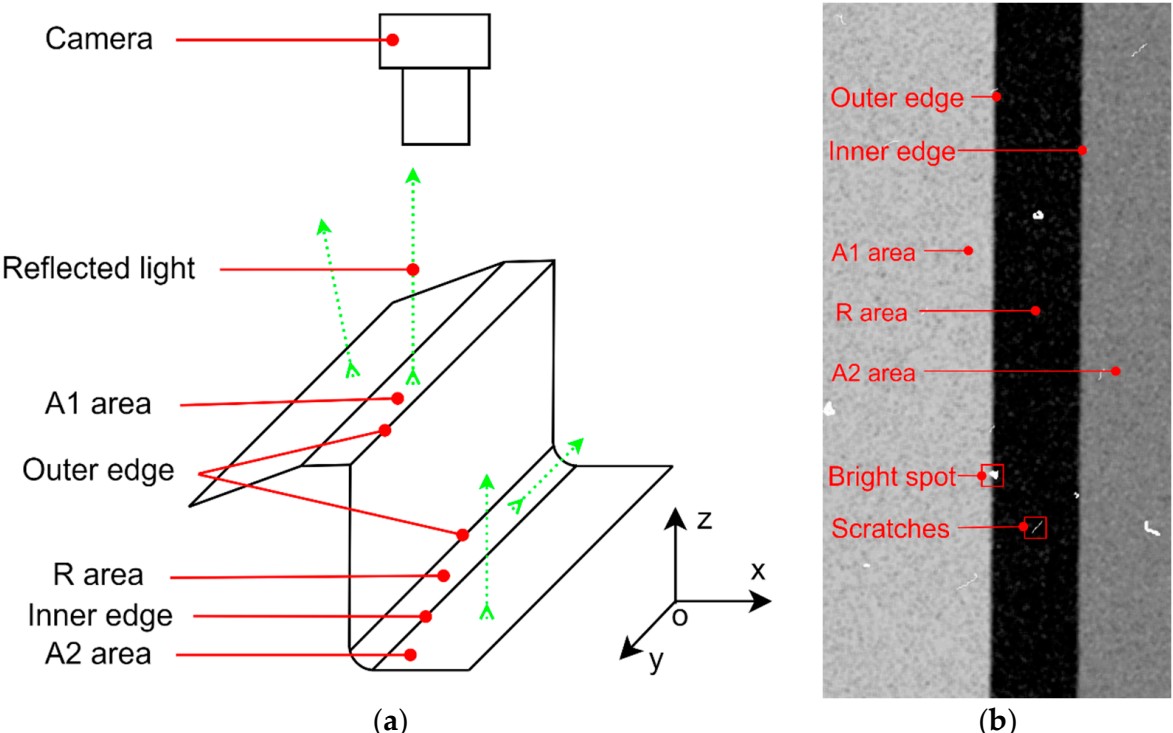

**Figure 6.** Surface imaging of the R region. (**a**) Surface structure and reflected light characteristics of the flat back cover. (**b**) Simulated local image of R area.

Distinct sections of the flat back cover exhibit distinct gray distributions when imaged as a result of differences in surface-reflected light, and the regional border is visible as an edge feature in the image [20]. On this basis, the local image of the R region is qualitatively simulated, as shown in Figure 6b. It can be seen from the analysis that the gray level of the regions on both sides of the outer edge changes the most, and the outer edge line is the longest line in the areas on both sides. In this way, this paper proposes a line detection method for the outer edge of the R region: (1) Extract and segment the outer edge area. (2) Adaptive line detection for external edges.

### 3.2. Region Extraction and Segmentation Based on Block Statistics

To accurately and quickly extract and segment the outer edge region, the image is divided into blocks, and the maximum standard deviation and mean value are calculated. The outer edge is roughly located according to the statistical standard deviation after which the area of the outer edge is extracted using the smallest bounding rectangle. The statistical mean value is used to guide the adaptive threshold segmentation. The description of Algorithm 1 is as follows:

| **Algorithm 1** Region extraction and segmentation based on block statistics |
| --- |
| **Input:** An image matrix **I**, the part of the object L, the sub-block width d, sub-block number k, |
| **Output:** An image matrix **O** |

| | |
| --- | --- |
| (1) | Initialization of parameters; |
| (2) | **O** = **I**.gray(L).ROI(L); // Graying the original image and get an area of interest |
| (3) | Compute Ld according to **O**, L, d, k; // The equipartition spacing of sub-blocks |
| (4) | **for** (i = 0 to k) **do** |
| (5) | Compute xi, yi according to i, Ld, k; // Position coordinates of sub-blocks |
| (6) | B = **O**.Rect(xi, yi, d); // Get sub-blocks image |
| (7) | **for** (j = 0 to **B**.cols or **B**.rows) **do** |
| (8) | R = B.Rect(j, ds); // statistic region |
| (9) | Compute S, M according to Equation (2); // block statistics |
| (10) | **if** (S > Sm) **then** |
| (11) | Sm = S; // Record maximum standard deviation |
| (12) | $T_i$ = M; // Save the average gray level |
| (13) | xi = j or yi = j according to L; // Record one coordinate |
| (14) | Compute xi or yi according to L; // Record the other coordinate |
| (15) | Record $Q_i(x_i, y_i)$; //Save the midpoint of the outer edge of the sub-block |
| (16) | Record $T_i$; |
| (17) | Compute $Q_a$ according to $Q_i$; // Get the midpoint of the outer edge |
| (18) | Compute T according to Equation (3); // Adaptive threshold |
| (19) | Compute $\theta_a$ according to $Q_a$ and $Q_i$; // the average slope relative to $Q_a$ |
| (20) | Compute $\beta_l$, $\beta_r$ according to $\theta_a$; // the angle range |
| (21) | **O** = **O**.Rect($Q_a$, $\beta_l$, $\beta_r$) .Binary(T); // Region extraction and segmentation |
| (22) | **return O**; |

The specific process of the algorithm is as follows:

(1) A region of interest (ROI) is established after graying the original image when an original image is obtained from the camera.

(2) In the ROI image, *k* sub-blocks with a width of *d* are divided symmetrically and equidistantly. Let the width of the rectangular statistics window in the sub-block be $d_s$. Equation (2) can be used to calculate the statistical window's average value *M(j)* and standard deviation *S(j)* for the sub-block $B_i$ when the statistical window position is *j* and the window area image is *R(j)*:

$$M(j) = \sum R(j)/(d_s \times d)$$
$$S(j) = \sqrt{\sum (R(j) - M(j))^2/(d_s \times d)} \tag{2}$$

(3) When A1 and R areas occupy the same area in the statistical window, the statistical window is located at the midpoint Qi(xi, yi) of the outer edge of the sub-block Bi, and the standard deviation of the window region is at its highest value, as shown in Figure 7. According to the maximum standard deviation, the segmentation threshold Ti of the center point Qi(xi, yi) of the outer edge in the sub-block Bi and the outer edge area is obtained.

(4) The midpoint $Q_a$ of the outer edge is roughly located with the help of the midpoint of the outer edge in *k* sub-blocks, and then the average slope of $Q_a$ is computed to roughly locate the angle $\theta_a$ of the outer edge. For the segmentation threshold $T_i$ of the outer edge area in *k* sub-blocks, the adaptive segmentation threshold *T* of the outer edge area is derived using Equation (3):

$$T = \sum_{i=0}^{k-1} T_i/k \tag{3}$$

(5) The precise angle $\theta$ of the outer edge is within the neighborhood $\delta$ of $\theta_a$. According to this, the angle range of the outer edge in the Hough polar coordinate system is

determined as $[\beta_l, \beta_r]$. The minimum bounding rectangle is constructed according to the angle range to extract the outer edge area. The collected region is then binarily segmented using an adaptive segmentation threshold $T$.

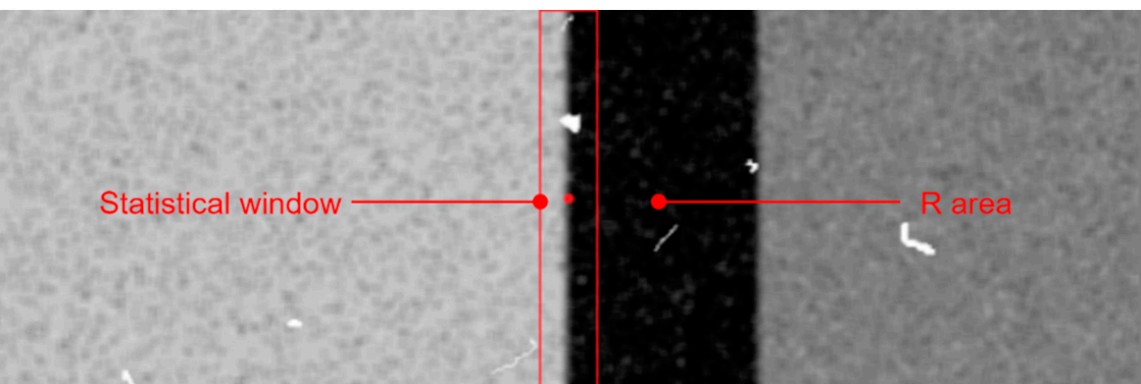

**Figure 7.** Position of maximum standard deviation.

### 3.3. Adaptive Hough Line Detection Based on Dichotomy

The Hough transform is chosen for line detection because it has the best identification ability and accuracy in an interference environment. The detection process is iterated by the dichotomy method because it has stable convergence and a quick turnaround time. The accumulator threshold corresponding to the longest line is iteratively searched based on the characteristic that the line at the outer edge of the R region is the longest. This enables the adaptive and high-precision line detection of the outer edge of the R region by steadily obtaining the optimal accumulator threshold. The description of Algorithm 2 is as follows:

| **Algorithm 2** Adaptive Hough line detection based on dichotomy |
|---|
| **Input:** An image matrix **I,** Hough transform discretization angle $\theta_h = 0.01°$ |
| **Output:** Information on a straight line (r, θ) |

| | |
|---|---|
| (1) | Initialization of parameters; |
| (2) | O = I.Canny(); //Edge extraction |
| (3) | **while** (F is true) **do** |
| (4) |   **if** $(v_r - v_l <= \varepsilon)$ **then** |
| (5) |     F = false; |
| (6) |     v = $v_l$; |
| (7) |   **else then** |
| (8) |     v = $(v_r - v_l)/2$; |
| (9) |   HoughLines(O, HL, $\theta_h$, $r_h$, v, $\beta_l$, $\beta_r$); // Hough line detection |
| (10) |   H = HL.size(); //Number of detected lines |
| (11) |   **if** (H > 1) **then** |
| (12) |     $v_l$ = v; |
| (13) |   **else if** (H < 1) **then** |
| (14) |     $v_r$ = v; |
| (15) |   **else then** |
| (16) |     F = false; |
| (17) | H = HL.size();//Number of detected lines |
| (18) | **for** (I = 0 to H) **do** |
| (19) |   r + = ri; |
| (20) |   θ + = θi; |
| (21) | r = r/H; // Get the average value |
| (22) | θ = θ/H; // Get the average value |
| (23) | **return** (r, θ)**;** |

The specific process of the algorithm is as follows:

(1) Canny operator is used to accurately extract the edge pixels from the outer edge after binary segmentation.

(2) The initial dichotomy interval is set as $[v_l, v_r]$, the length of convergence interval as $\varepsilon$, the distance and angle after Hough transform discretization as $r_h$ and $\theta_h$, respectively, and the angle range of line detection as $[\beta_l, \beta_r]$. The value of the accumulator threshold $v$ is set to $v_l$ and the iteration flag $F$ is set to be true. According to the iteration flag $F$, it is a cyclic process to determine whether to perform a binary search. A loop procedure that uses the iteration flag F to determine whether to conduct a dichotomous search is employed.

(3) When $v_r - v_l > \varepsilon$ is true, the accumulator threshold $v$ should be set to $(v_r + v_l)/2$; when $v_r - v_l <= \varepsilon$ is true, the accumulator threshold $v$ should be set to $v_l$, and $F$ should be set to false. According to the accumulator threshold, the Hough transform is used to detect the edge, and the distance and angle of $H$ lines are obtained.

(4) Determine the number of lines: when $H > 1$ is true, let the left end point of the second partition $v_l = v$; when $H == 0$ is true, let the right end point of the second partition $v_r = v$; and when $H == 1$ is true, the cycle should finish.

(5) The final straight line detection findings are obtained at the cycle's conclusion. If there are multiple approximately equal solutions $(r, \theta)$ in the detection, the multiple results would be averaged to generate the unique $(r, \theta)$.

The properties of the ideal line serve as a representation of the outer edge of the R region when the aforementioned iterative search procedure is complete, and the midpoint of the optimal line is regarded as the midpoint of the outer edge. The midpoint coordinates of the line in the local image of the R region can be obtained according to the algebraic relationship. The coordinates of the midpoint as the final result of the edge detection outside the R region will be saved in the disk file as the input value for the full-size measurement.

## 4. Experiment and Analysis

### 4.1. Verification of the Algorithm using Key Parameters

The key algorithm parameters in this paper are sub-block width $d$ = 381, sub-block number $k$ = 13, and the Hough transform discretization angle $\theta_h$ = 0.01°. When the following experimental verification is performed, the CPU frequency is 2.4 GHz and the running memory is 4 GB on a laptop computer.

The outer border straight lines are recognized under various $d$ values as the test data, and 320 correctly detected local images in the R region are used as the supervision data to confirm the accuracy of region extraction and segmentation. As illustrated in Figure 8, the influence of the sub-block width $d$ on the error rate is achieved when the detection results show that the midpoint error of the straight line surpasses 2 pixels. The results show that the error rate can be reduced by increasing the sub-block width $d$, and value that is too large would reduce the accuracy of pixel statistics. The maximum error rate is 0.3125% when $311 < d < 441$. The accuracy of region extraction and segmentation in this algorithm is as high as 99.68%. The statistical region's statistics are prone to blemishes and scratches when $d$ is too small. The properties of the target edge in the statistical window will be diminished when $d$ is too large. The measurement error will rise in the two aforementioned situations.

Different $k$ values were tested to evaluate the average time consumption of region extraction and segmentation in 16 photos as well as the average time consumption of the algorithm in order to confirm the efficiency of region extraction and segmentation. Figure 9 displays the data and demonstrates that the time required for region extraction and segmentation increased linearly as the $k$ value rose. When the $k$ value is relatively small, the proportion of region extraction and segmentation in the total time of the algorithm is small, allowing the algorithm to spend the majority of its time generating high-precision iterative values. The time consumption of region extraction and segmentation in this paper was less than 60 ms, and the total processing time was less than 200 ms.

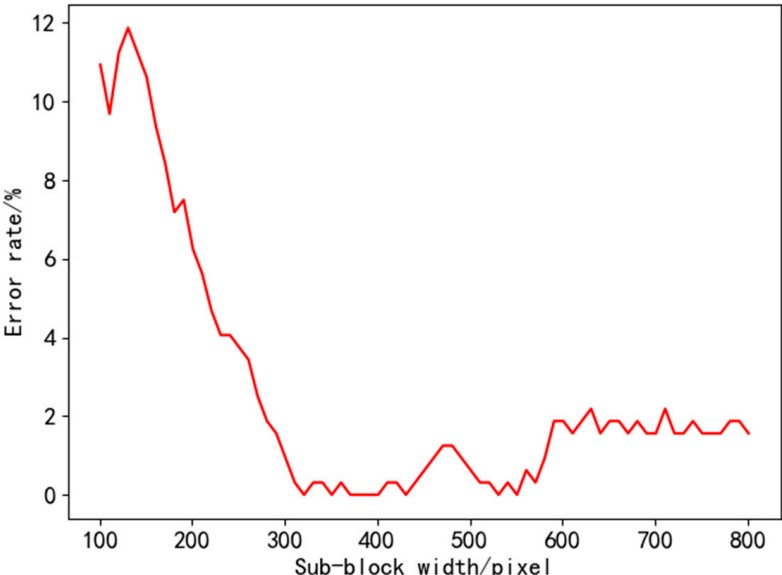

**Figure 8.** Effect of sub-block width on error rate.

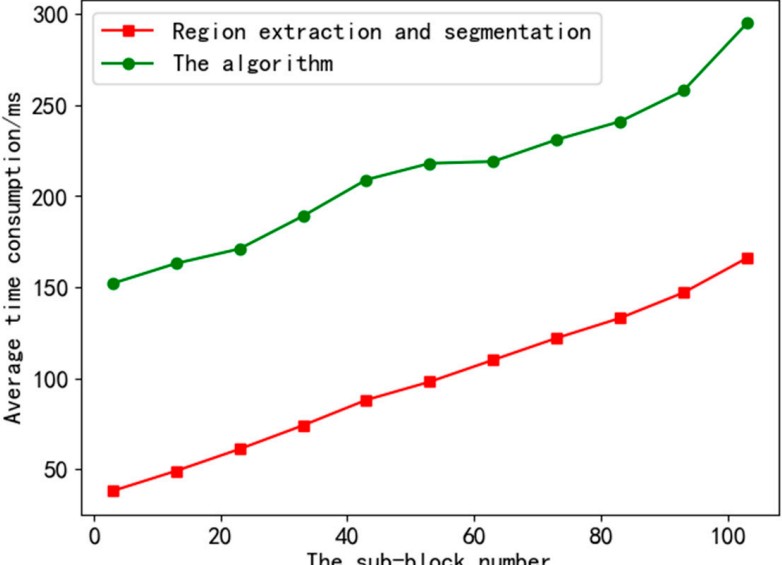

**Figure 9.** The influence of the sub-block number on the average time consumption of the algorithm.

To verify the high accuracy of adaptive line detection, this paper quantitatively simulated the local image of the R region according to the theoretical analysis of surface imaging and image characteristics. The position and angle of the outer edge of the image were known, and rotation transformation was used to derive theoretical values for various angles and positions of the outer edge. The algorithm in this paper was used to detect the external edge to obtain the detection value and compare the angle and position errors of the external edge's linear detection. The outcomes are displayed in Figure 10, which demonstrates that the angle error of the outer edge line detection was less than 0.01°, and the position error was less than 0.2 pixels. Of course, the smaller the $\theta_h$ value, the higher the accuracy of line detection and the smaller the detection error of the algorithm to the edge, but the execution speed of the algorithm will greatly increase.

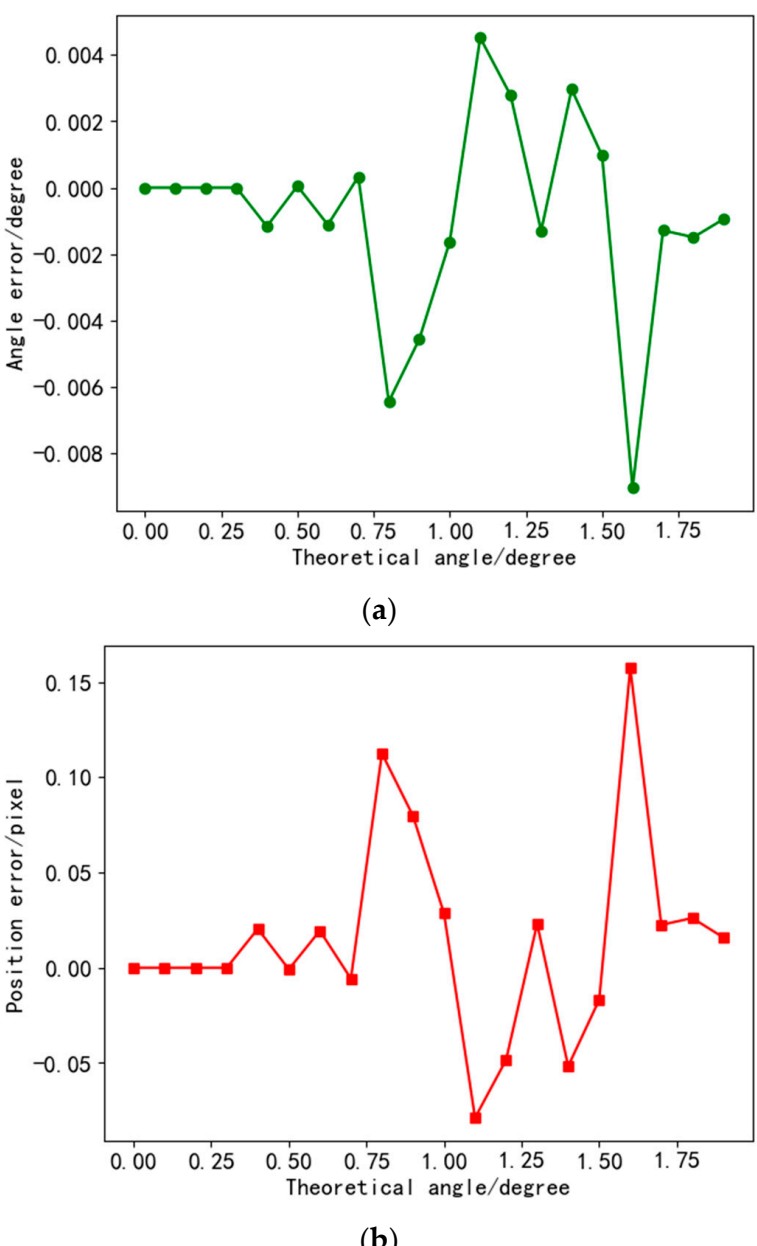

**Figure 10.** Errors of the outer edge linear detection. (**a**) Detection angle error. (**b**) Detection position error.

*4.2. Verification of the Algorithm Detection Effect*

To verify the optimal detection effect of the algorithm in this paper, after the extraction and segmentation of the outer edge of the local image in the actual R region, LSD line detection, Hough line detection, LS line fitting and the algorithm in this paper were each used to detect the outer edges, respectively. Figure 11 displays the results of various algorithms' line detections on the outer border. The findings demonstrate that the LSD algorithm was unable to discern continuous lines when the outer edges were damaged due to edge breaking and blurring caused by surface flaws and an uneven gray scale. With Hough line detection, it was difficult to set the optimal accumulator threshold to detect the outer edges. LS line fitting deviated from the outer edges due to the interference of outliers. The algorithm in this paper can withstand interference and accurately detect the line on the outer edge.

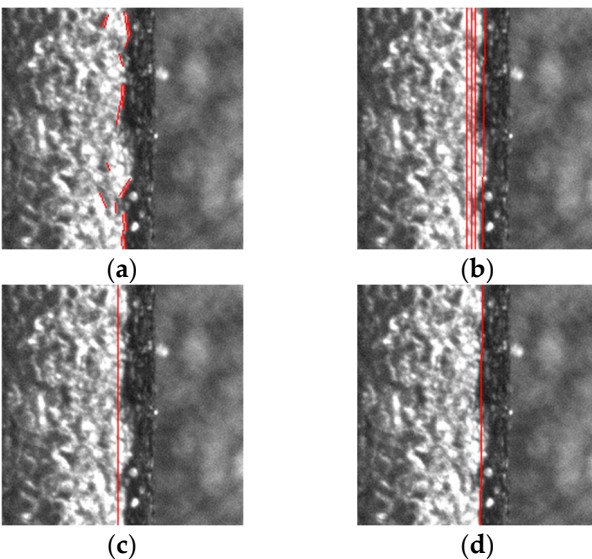

**Figure 11.** Detection results of different line detection algorithms. (**a**) LSD line detection. (**b**) Hough line detection. (**c**) LS line fitting. (**d**) The algorithm in this paper.

The linear detection results of the outside edge were achieved by detecting the local images of the R region in various actually collected portions, as shown in Figure 12, to confirm the applicability of the algorithm in this study. The findings demonstrate that the algorithm is capable of properly and adaptively identifying the exterior R region edges that match local images of various components, realizing the entire size measurement of the R region of the flat back cover.

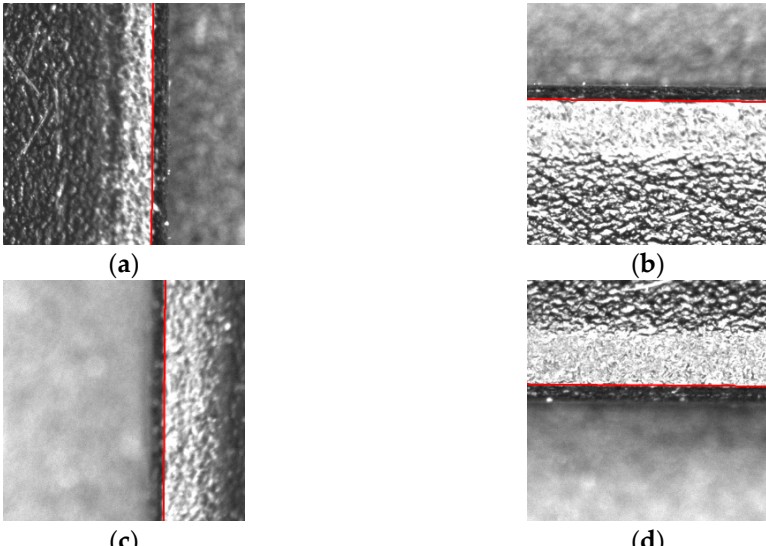

**Figure 12.** Straight line detection results of outer edges at different parts. (**a**) Schematic diagram of *L*1 test results. (**b**) Schematic diagram of *L*2 test results. (**c**) Schematic diagram of *L*3 test results. (**d**) Schematic diagram of *L*4 test results.

### 4.3. System Error Analysis

By identifying the angular points on the calibration plate of the plane using Zhang's calibration method [21], the internal and exterior parameters of the camera were determined in order to establish the mapping from the pixel coordinate system to the physical coordinate system. The standard distance of the grid in the calibration plate of this paper was 2.5 mm, and the measured value of the CCD camera was 2.50193 mm. Thus, the calibration error was under 0.002 mm.

Since the error in the moving distance of the mobile system was inevitable, this paper has realized the error compensation of the mobile system by obtaining the moving distance fed back by the grating ruler many times. The distance from the outer edge of the R region at *L*1 and *L*3 was defined as the length of the R region and at *L*2 and *L*4 as the width of the R region. A total of 20 groups of flat back covers were measured in full size using the technique and measuring platform presented in this research. The measured values were collected within 6 s and compared with the manually recorded values to obtain the size measurement error in the R region, as shown in Figure 13. The results show that the measurement error of the measuring system is less than 0.03 mm.

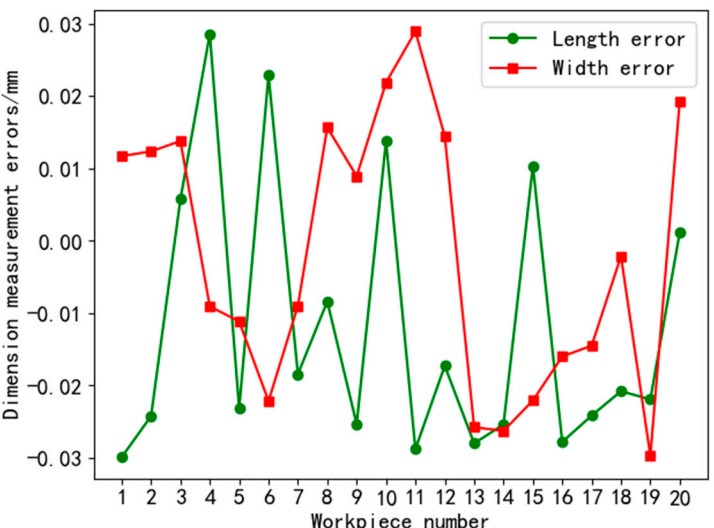

**Figure 13.** Dimension measurement errors in the R region.

The same batch of samples from the actual project were measured every other day to check for stability, and the findings revealed that the system error stayed within 0.03 mm. The examination of system errors reveals that even while inadvertent errors are challenging to remove, system errors are stable, minimal, and fulfill the project's genuine requirements.

## 5. Discussion

In this study, monocular vision was integrated with a linear motor for precision positioning, extending the working range of the vision system's field of view to include all linear motor strokes. The system described in this work can theoretically measure the precision size of things with a maximum size of $340 \times 340$ mm, solving the issue of precision measurement of large-size objects in earlier studies. This system can theoretically measure the precision size of objects with a maximum size of $340 \times 340$ mm. The simplicity of usage will be lessened because the system developed in this work is also rather sophisticated and requires important skills in the disciplines of control, image algorithms, communication, programming, etc.

According to the experimental results, the sub-block width *d* is a crucial parameter that, when the value is incorrect or inappropriate, greatly affects measurement accuracy. From a statistical perspective, *d* is not only a singular value; all values within a given interval are valid. However, the value used in this paper is biased toward the middle of the interval so that *d* can be more reliably used to minimize measurement error. The number of sub-blocks *k* is more likely to influence how quickly the algorithm executes, and *k* can be chosen more carefully to increase efficiency while still satisfying usability. The accuracy of line detection and the duration of dichotomy iteration for edge line detection in this work will be impacted by the discretization angle, which has a significant impact on the final detection effect. The algorithm also makes mistakes when trying to find straight lines. In this study, the discretization angle can match the straight line detection accuracy requirements at $10^{-2}$ levels while also taking measurement efficiency into account.

The measuring instrument's measured value is utilized as the true value in the system error experiment, although it can only be used as an approximation of the genuine value. Because the true system error is difficult to collect and contains numerous elements, the calculated system error also represents the real error. In actuality, we focus more on the stability and fluctuation range of the system fault. The experimental findings demonstrate that while the system error has always existed, it has remained within a narrow range that may be measured for use in practical tasks.

## 6. Conclusions

(1)  To realize the precise measurement of a screen gap, a mobile vision measurement system was designed. A total of 16 local image acquisition positions were set in the rounded area of the flat back cover to obtain the information of local outer edge. Combined with the positioning information, the local information was fused to realize the full-size measurement in the R region of the flat back cover, and, finally, the precise size measurement of the flat screen gap was realized.

(2)  An adaptive line detection method for the outer edge of the R region was proposed. The accuracy of extracting and segmenting the outer edge area was up to 99.68%. Time consumption was less than 60 ms. The angle error of line detection was less than $0.01°$, and the position error was less than 0.2 pixels. It had optimal anti-interference performance and could realize the adaptive and accurate positioning of the outer edge of the R region within 200 ms. Under the algorithm and measurement system in this paper, the measurement error of the R region size of the flat back cover was less than 0.03 mm, and the measurement time was less than 6.0 s.

In the meantime, the system and methodology used in this work could potentially be improved. Linear motors have repeatable positioning precision, and their positioning error has a significant impact on the system measurement's accuracy. Therefore, it is necessary to research ways to lower linear motor positioning errors. The establishment of a mathematical model for system error has essential guiding importance for system improvement even though there is no theoretical analysis of numerous system error factors in this paper. The technique suggested in this study has a good detection impact on objects with similar edge features, but it is still necessary to confirm the algorithm's detection effect on objects with different edge feature types. The algorithm's adaptability can be confirmed and enhanced in the future. The mechanism envisioned in this paper is not inexpensive. A linear motor is more expensive because it is a novel product. The mobile vision system created in this research will have a wide range of applications as the manufacturing sector develops in the future. With strong flexibility, the mobile vision system and full-size measuring method described in this study can also be applied to the domains of visual positioning, defect detection, and the pattern identification of large-size items.

**Author Contributions:** X.Y. and F.W. completed the research and contributed to the writing of the manuscript. Q.Y. and X.H. conceived the project and provided the research methodology. J.M. and L.S. conducted the formal analysis and project administration. All authors have read and agreed to the published version of the manuscript.

**Funding:** This research was supported by the National Natural Science Foundation of China (No. 61976083) and the Hubei Province Key R&D Program of China (No. 2022BBA0016).

**Institutional Review Board Statement:** Not applicable.

**Informed Consent Statement:** Not applicable.

**Data Availability Statement:** The code and datasets involved in this study are publicly accessible and downloadable at https://github.com/InsomniaMonster/MobileVisionData1.git (accessed on 8 May 2023).

**Acknowledgments:** The authors would like to thank Xuhui Ye for his early proposal for this manuscript.

**Conflicts of Interest:** The authors declare no conflict of interest.

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
