# Peer review of "Research on the Precision Measurement Method of Flat Screen Gap Based on Mobile Vision"

_applsci, doi:10.3390/app13126909_

Round 1

Reviewer 1 Report

1- Readability of the paper is hard.

2- There is no figure (1). Figures start by figure (2)

3- The main idea needs more clarification in the introduction. 

4- There is a subtitle (2.1. Structure design of mobile vision measurement system) which has no text content.

5- The operation algorithms in sections (2, 3) are missing.

6- A comparison with the previous research is not given.

The paper readability is hard. It should be easy for the readers (specially researchers) to get the main idea and to go through the paper methodology to be able to benefit from the scientific content.

Reviewer 2 Report

The study is well performed. The statistical analysis is accurate. The results are clearly and fairly reported. The discussion is relevant. In my opinion, it can be published in this journal. Useful careful evaluation of syntax

Useful careful evaluation of syntax

Reviewer 3 Report

I recommend processing the following comments:

-       1.  Resources are missing in the figures.

a    2 add text under the figure number 13 and more to describe the individual parameters of the dimension of the measured errors, the chapter cannot end with the figure 

-      3.   Add a discussion section to discuss results and findings with other studies on the topic.

-        4. The conclusion missing the future direction.

-       5.  English check should be required.

The article contains minimal grammatical errors which should be removed even though they do not reduce the quality of the article.
